# Network Model for Online News Media Landscape in Twitter

**Ford Lumban Gaol [1],* , Tokuro Matsuo [2] and Ardian Maulana [3]**

[1]   Computer Science Department, Bina Nusantara University, Jakarta 11480, Indonesia
[2]   Information System Department, Advanced Institute of Industrial Technology, Tokyo 140-0011, Japan
[3]   Information Systems Management Department, Bina Nusantara University, Jakarta 11480, Indonesia
*   Correspondence: fgaol@binus.edu

**Abstract:** Today, most studies of audience networks analyze the landscape of the news media on the web. However, media ecology has been drastically reconfigured by the emergence of social media. In this study, we use Twitter follower data to build an online news media network that represents the pattern of news consumption in Twitter. This study adopted a weighted network model proposed by Mukerjee et al. and implemented the Filter Disparity Method suggested by Majó-Vázquez et al. to identify the most significant overlaps in the network. The implementation result on news media outlets data in three countries, namely Indonesia, Malaysia, and Singapore, shows that network analysis of follower overlap data can offer relevant insights about media diet and the way readers navigate various news sources available on social media.

**Keywords:** network analysis; network model; online news media; media network; twitter

## 1. Introduction

Global trends show the increasing tendency of people to get news through social media. Reuters Institute Digital News Report 2017 reports that around 63% of public gets news from social media [1]. Social media platforms increasingly mediate the relationship between readers and publishers. Social media helps the public to know the latest trends faster, as well as aids news media outlets to reach their audiences widely [2–4].

This development shows that social media platforms have changed the way we access information and form opinions. Moreover, the emergence of various new phenomena in the realm of social media, such as opinion polarization and echo chamber, is closely related to how the public consumes news [5–9]. This situation raises the need to better understand news audience formation in social media environments.

On social media, users interact with news sources through commenting and sharing the news published by the news outlet account. In an information-rich environment, these interactions collectively contain important information about how the public consumes news: user preferences are reflected through the news outlets they follow [6–8], while the presence of shared followers between news outlets indicates a common readership base that collectively forms a pattern of interconnection between news media [9].

In this study, we use a network approach to map the news media landscape in one of the most popular social media platforms, Twitter. Twitter is useful for studying the media landscape because of the large number of users, open access to data, and most importantly all interactions from both the news media and the audience recorded online [10]. So far, Twitter has become a laboratory of various studies related to the dynamics and propagation of information through social networks [11,12].

More specifically, this study proposes a methodology to build and analyze the network of online news media in Twitter based on follower overlap data. As shown in previous studies of audience

networks [9,10,13–24], audience overlap among news sources offer important insights on how the audiences navigate through many news sources that are available online. In the context of the Twitter environment, follower overlap between news media accounts can be interpreted as follows. First, as a measure of similarity: the more followers any two news media accounts share, the closer those news outlets are in terms of their audience base. Second, as a measure of the diversity of people consuming a given news outlet: if a media outlet has shared followers with many other media, it means that the followers of that media outlet have a wide range of interests.

Our proposed methodology adopts a weighted network model proposed in [19] and extends its application to the behavior of news consumption in social media environment. However, in contrast to [19], in this study we employ the Filter Disparity Method [25], as used in [21], to identify the most significant overlaps in the network. In this study we demonstrate how to apply the methodology we proposed by examining the news media landscape in three countries, namely Indonesia, Malaysia, and Singapore.

In general, considering the development of social media as the main news distribution platform, this research contributes to enrich the literature on empirical studies of news consumption behavior in the social media environment. Specifically, this study makes three contributions. First, this study proposes a framework for building and analyzing the news media landscape in Twitter using follower overlap data. Second, this study explores the characteristics of news outlets on Twitter and confirms the findings from previous studies regarding the distribution pattern of audience size. Third, this study measures the structural position of news outlets and identifies the prominent media in the news media landscape in three countries analyzed. The analysis shows that audience size of a media outlet does not always reflect the centrality of the outlet position in media network.

The remainder of this paper is organized as follows: news consumption studies using a network approach are reviewed in Section 2. The steps for implementing a network approach to news media follower overlap data is proposed and explained in Section 3. In Section 4 we will report the implementation results of the proposed methodology by using news media data in three countries, namely Indonesia, Malaysia, and Singapore. In Section 5, we will discuss the relevance of the network approach in providing new insight about the news media landscape in social media space. Section 6 concludes this paper.

## 2. Related Work

The concept of audience overlap has a relatively long history in media studies [11], but the potential of overlap data to build a network of media was only lately uncovered [9,10,12–22]. Network is a natural representation of the news media landscape. In addition, network analysis provides analytical tools needed to investigate a number of issues in media studies such as fragmentation in news market, patterns of media clustering, media centrality, and structural comparison of media networks between countries.

Reference [14] is the first study that proposes analyzing audience duplication data as a network, where the nodes represents media outlets and the level of duplication between their audiences are represented by ties or edges. A number of other studies that used the same methodology soon followed this work [15–18]. However, these studies disregard a number of important things, such as edge weight that represents the strength of the overlap, and there was no assessment of the statistical significance of the observed overlap. More recent research has proposed some methodological improvements to the original approach [9,10,19–24].

The pattern of relationship in media networks depends on how we measure the relationship between media nodes. Mukerjee et al. [19] proposed the phi coefficient as a metric to measure the strength of audience overlap between news outlets. Unlike the metrics used in previous studies [14], the phi coefficient not only considers the size of the overlapping audience between news outlets, but also the relative size of the audience of each outlet. In addition, Mukerjee et al. [19] also applied a statistical test, namely the *t*-test, to filter out insignificant overlaps between media outlets. With

this approach, Mukerjee et al. was able to draw a convincing conclusion regarding the structural characteristics of the analyzed media networks.

Majó-Vázquez et al. [21] made a methodological contribution by using the Disparity Filter algorithm to identify the most significant overlap in audience networks. The Disparity Filter algorithm is a network reduction technique developed by Serrano et al. [25] to identify the 'backbone' structure of a weighted network. As opposed to a static value of the *t*-test used in [19], this algorithm operates the null model to accept or reject edges based on the distribution of their weights at the node (egocentric) level. In [21] Majó-Vázquez et al. show that the implementation of this method can sufficiently reduce the network density without destroying the multi-scale nature of the network.

However, most of the studies mentioned above use web traffic data to build audience networks. We highlight the work of An et al. [10] and Hahn et al. [24] who used a network approach to explore the news media landscape in Twitter. An et al. [10] uses a directed weighted network to represent news media landscape in Twitter, and applies a closeness metric, which is the probability that random followers of media B also follow media A, to measure the closeness value of media A from media B. For all directed pairs of media sources, this study calculates the closeness metric of media A to all other media sources and examined which ones appear the closest. However, compared to the Disparity Filter method, the network reduction technique used in [10] has limitations because it ignores the multi-scale nature of media networks. In contrast to [10], Hahn et al. [24] uses the symmetry form of the closeness metric to measure similarity between news outlets, and then they represent the news media landscape as a weighted undirected network. Furthermore, Hahn et al. [24] did not make any effort to reduce the density of the resulting media network.

In the same spirit as [10] and [24], in this study we use the phi coefficient to measure the similarity between two news outlets based on the number of common followers. We will further apply the Disparity Filter method, as used in [21], to eliminate insignificant edges as well as to sparsify the resulting media networks. In general, we propose a framework for applying a network approach on Twitter follower data, from data collection strategies to network analysis using node centrality indicators.

## 3. Proposed Methodology

The research methodology consists of three stages, namely data extraction, modeling and simulation, and analysis, as shown in Figure 1. Based on this methodology, this research will process online news media data in Indonesia, Malaysia, and Singapore, then represent the data as a media network, and measure a number of network statistics that depict the online news media landscape in the three countries analyzed.

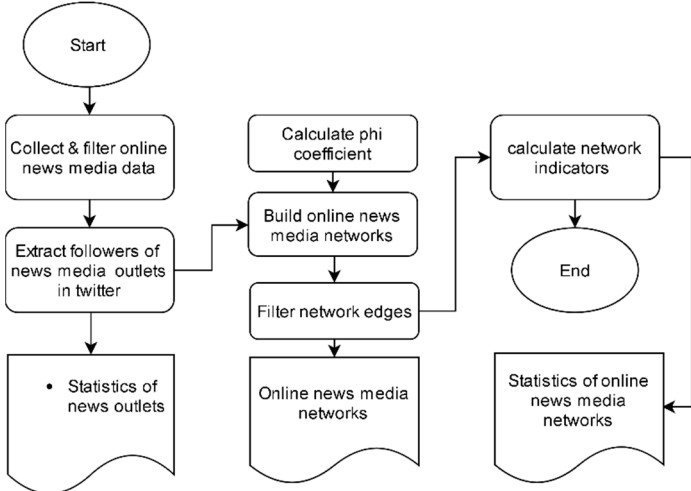

**Figure 1.** Research methodology

*3.1. Data Extraction*

Firstly, in the data extraction stage, researchers collected a list of online news-media outlets along with their Twitter account. These media outlets were selected based on two criteria, as follows: (i) the outlet has an active Twitter account. Online news media usually have an official Twitter account where social media users can follow to get news updates; (ii) the Twitter account of the news media outlet has a significant number of followers. This study uses a threshold value of 0.007 percent of the largest number of followers of news media accounts in each country. The media selection process is implemented for filtering news media outlets that are not relevant for analysis [18].

Based on the collected media list, the process of extracting followers was then carried out to retrieve followers of each of these media accounts on Twitter. In this study, the follower extraction process used Twarc, the Python package for Twitter data archiving [26]. This process is very time consuming considering the number of followers of media outlet can reach millions of users. The main constraint is the Twitter rate limit of 18,000 accounts per 15 min. For example, the extraction process of approximately 15 million followers of the detik.com account required 139.9 h or about 6 days. Considering these constraints, we only stored the information needed such as Twitter id of the media outlets and Twitter id of the media followers. The final step in this stage was to do data cleaning. The pseudo code for the follower extraction process is shown in Algorithm 1.

---

**Algorithm 1** follower extraction

---

1: **Begin**
2: **Input**:
3:　　U: list of online news outlets;
4:　　fmin: minimum number of followers of a news outlet;
5:　　dmax: maximum day of inactivity;
6:　　d: today date;
7:　　n: number of all news outlets
8: **Process**:
9:　　A ← ∅;
10:　　**for** i =1 to n:
11:　　　　dlast ← extract date of outlet i latest tweet;
12:　　　　f ← extract number of outlet i followers;
13:　　　　**if** d - dlast < dmax:
14:　　　　　　**if** number of outlet i followers > fmin:
15:　　　　　　　　A(i) ← collect all user ids that follow outlet i;
16:　　　　　　**else**:
17:　　　　　　　　continue;
18:　　　　　　**end if**
19:　　　　**else**:
20:　　　　　　continue;
21:　　　　**end if**
22:　　**end for**
23: **Output**: A: list of followers of all news outlet
24: **End**

---

*3.2. Modeling and Simulation*

The second stage consists of three main activities, namely the measurement of similarity between news outlets, the construction of networks, and edge filtering.

3.2.1. Similarity Measurement

The first step was to measure the level of similarity between any two news outlets based on audience preferences reflected in their followers' data. Intuitively, this measurement depends on two things, namely the reader preferences for both media (i.e., whether the reader follows/or does not

follow the two media), and the preferences of the reader that are specific to each media. In this study, following [19], the measurement of similarity was framed as a correlation between two binary variables using the phi coefficient. The phi coefficient is an index of the relation between any two sets of scores that can both be represented on ordered binary dimensions, i.e., follow or not follow. For two random variables x and y we have a 2 × 2 table contingency table as in Table 1.

**Table 1.** Contingency table.

|        | y = 1       | y = 0       | Total      |
| ------ | ----------- | ----------- | ---------- |
| x = 1  | $n_{11}$    | $n_{10}$    | $n_{1.}$   |
| x = 0  | $n_{01}$    | $n_{00}$    | $n_{0.}$   |
| Total  | $n_{.1}$    | $n_{.0}$    | $n$        |

Where $n_{11}$ is the number of followers overlap of x and y, $n_{10}$ is the number of followers of x that do not follow y, $n_{01}$ is the number of followers of y that do not follow x, $n_{00}$ is the number of population elements that are not followers of x and y, $n_{1.}$ is the number follower of x, $n_{.1}$ is the number of followers of y. All table elements are non-negative counts of numbers of observations that sum to n, the total number of followers. The phi coefficient that describes the association of x and y is most easily computed as,

$$\varphi_{xy} = \frac{n n_{11} - n_{1.} n_{.1}}{\sqrt{n_{1.} n_{.1} (n - n_{1.})(n - n_{.1})}}. \tag{1}$$

The phi coefficient ranges from −1 to +1, with negative numbers representing negative relationships, zero representing no relationship, and positive numbers representing positive relationship. Two news outlets are considered positively associated if most of the data falls along the diagonal cells. In contrast, two news outlets are considered negatively associated if most of the data falls off the diagonal. In other words, the more followers of two news outlets overlapping relative to their complement, the more positive the value of $\varphi_{xy}$, and the more similar the two outlets are in terms of their audience base.

Algorithm 2 shows the pseudo code for measuring news outlets' similarity. For each pair of news outlets, we take followers of both media and calculate the value of the phi coefficient using Equation (1). The measurement results are stored in the form of n × n association matrix of news outlets.

---

**Algorithm 2** similarity measurement

---

1: **Begin**
2: **Input**:
3:    A: list of followers of all news outlet;
4:    n: number of news outlets;
5:    Let C be a nxn association matrix of news outlets;
6: **Process**:
7:   **for** i=1 to n
8:      $A_i$ ← retrive followers of outlet i;
9:     **for** j=1 to n:
10:       **if** i =/= j:
11:         $A_j$ ← retrive followers of outlet j;
12:         phi = calculate follower similarity between outlets i and j using equation(1);
13:         C(ij) ← phi;
14:       **else**:
15:         C(ij) ← 0
16:       **end if**
17:     **end for**
18:   **end for**
19: **Output**: C: nxn association matrix of news outlets;
20: **End**

---

### 3.2.2. The Construction of News Media Networks

News media networks is modeled as weighted graphs that represent relationships between media outlets based on their followers overlap. Formally the media network is defined as weighted graph G (V, E, W), which is composed of nodes $v_i$ and the edge $e_{ij}$, where the node of $v_i$ is the outlet $i$ in the news-media set $V$ ($v_i \in V$; $i = \{1, \ldots N\}$), and the edge $e_{ij}$ ($e_{ij} \in E$) represents the relation between outlets $i$ and $j$, with the weight of the relations $w_{ij}$ ($w_{ij} \in W$). The weight of the edge represents the strength of relationship between the two media outlets, which indicates the amount of audience the outlets share. In this study, phi coefficient ($\phi_{ij}$) is used as an indicator of the strength of relations between any two news outlets where outlets $i$ and $j$ will be connected by an edge if the phi coefficient of the two outlets is positive.

Based on the association matrix that was built previously, we identified all media pairs with positive elements and represented them as a weighted undirected network of news media. The pseudo code for network construction is shown in Algorithm 3 and implemented using NetworkX, a Python package for complex network studies [27].

---

**Algorithm 3** Construct news media networks

---

1: **Begin**
2: **Input**:
3:　　C: nxn assocation matrix of news outlets;
4:　　U: list of news outlets;
5: n: number of news outlets;
6:　　**Process**:
7:　　V ← Ø; E ← Ø; W ← 0;
8:　　V ← U; // all news outlets are vertices in network G;
9:　　**for** i=1 to n-1:
10:　　　　**for** j=i+1 to n:
11:　　　　　　**if** C(i,j)>0:
12:　　　　　　　　u = U(i);v = U(j); w(u,v)= C(i,j);
13:　　　　　　　　E ← E ∪ e(u,v);
14:　　　　　　　　W ← W ∪ w(u,v);
15:　　　　　　**else**:
16:　　　　　　　　continue;
17:　　　　　　**end if**
17:　　　　**end for**
18:　　**end for**
19: **Output**: G(V,E,W): news media network;
20: **End**

---

### 3.2.3. Edge Filtering

The next step is to filter out network edges that are not statistically significant [28]. As a correlational network, media networks constructed using Algorithm 3 will naturally have a dense structure. In addition, it is known from the previous studies [19,21] that media networks have very heterogeneous interaction patterns, with degree and weight distributions varying over many orders of magnitude. These features make filtering techniques such as global weighted threshold or *t*-test inappropriate for media networks because the methods overly simplify the multi-scale nature of media networks, destroy local cycles, and overlook nodes with small strength [25]. This study uses a Disparity Filter method to extract the relevant connection backbone in media networks, preserving the edges that represent statistically significant deviations with respect to a null model for the local assignment of weights to edges. The Disparity Filter algorithm operates in node level based on the following null hypothesis: the normalized weights that correspond to the connections of a certain node of degree k are produced by a random assignment from a uniform distribution [25]. Then, for every edge featured

in ego network of a media node, the algorithm tests its weight against the null model, and store it if it survives that test. Algorithm 4 shows the pseudo code for the edge filtering process where we used the python-backbone-network package [29] to extract the backbone of media networks.

| **Algorithm 4** edge filtering |
| --- |
| 1: **Begin** |
| 2: **Input**: |
| 3:　　G(V,E,W):news media networks; |
| 4:　　alpha: 0.005; |
| 5:　　n: number of news outlets; |
| 6: **Process**: |
| 7:　　G′ ← disparityfilter(G,alpha); // (Achananuparp,2013) |
| 8:　　V ← U; // all news outlets are vertices in network G; |
| 9: **Output**: G′(V,E,W): news media networks; |
| 10: **End** |

## 4. Results

In the early stages of data collection, this study succeeded in collecting 215 Indonesian online news media, 111 Malaysian online news media, and 58 Singaporean online news media. One of the main sources is a list of the most popular sites in a country published by Alexa [30]. This study utilized the Alexa rank, a site rating system based on web traffic data, to ensure that the analyzed media outlets have a significant sized audience. The data collected was then further selected using the two criteria described earlier. Based on the final list, the follower extraction process was then carried out to retrieve followers of the news outlets on Twitter.

The number of online news media varies in the three analyzed countries. As shown in Table 2, the number of news portals in Indonesia is far more than the other two countries, both before and after the selection process. This is not surprising given the size of Indonesia's population and the increasing of internet access, and most importantly, press freedom in Indonesia is better than Malaysia and Singapore [31].

**Table 2.** Number of online news media before and after edge removal.

| Country | Number of News Media before Selection | Threshold (0.007%) | Number of News Media after Selection |
| --- | --- | --- | --- |
| Indonesia | 215 | 1054.963 | 166 |
| Malaysia | 111 | 123.9864 | 86 |
| Singapore | 58 | 70.57435 | 42 |

The readership of news outlets in the three countries analyzed is also very diverse. As shown in Figure 2, the histogram of followers of news outlets on Twitter has a right-skewed shape with a long right tail. This means that there are only a handful of news outlets with a very large number of Twitter followers, while the majority of news outlets only have a relatively small number of followers. This indicates that, despite the variety of news media available online, most consumers look to the most prominent media outlets [15,32]. This distribution pattern was also found by previous studies that analyzed news consumption on different media platforms [9,19], which further confirms the classic long-tail argument in media research that online news sources are very heterogeneous in terms of the size of the audience [33,34].

Based on the follower data, this research builds news media networks where the edges tell us how many followers overlap between any two news sources on Twitter. Table 3 shows the statistics of news media networks for three countries analyzed. Before edge removal, the news media networks are very dense, that is, most news outlets share followers with most other outlets. As shown in Table 3, the

density of the initial networks, that is ratio of edges to total possible number of edges in the networks, almost reaches the maximum value, i.e., 0.912 for Indonesia, 0.839 for Malaysia, and 0.927 for Singapore. However, the difference in the number of followers shared, and in the total followers of each news media (popular outlets have a larger audience to share than smaller outlets) causes the strength of the overlap to vary across media pairs. Consequently, not all edges are significant. An edge is not statistically significant when the probability of observing a given overlap is very unlikely under the null hypothesis of random overlap distribution. Edge filtering step using the Disparity Filter method is carried out to eliminate edges that do not reach the significance level for a probability value $p < 0.05$. On average, about 94% of all original edges were successfully removed from the three networks analyzed, i.e., 11,592 edges on the Indonesian media network, 3441 edge on the Malaysian media network, and 752 edges on the Singapore media network. As a result, as shown in Table 3, the density values of the three analyzed networks are very small, indicating that the final networks have a sparse structure.

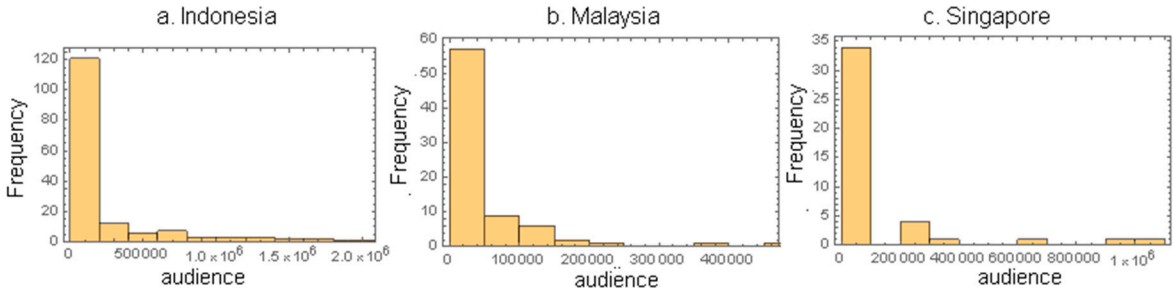

**Figure 2.** Histogram of Twitter followers for online news media outlets in Indonesia, Malaysia, and Singapore.

**Table 3.** Statistics for online news media networks before and after edges removal.

|  | Indonesia | | Malaysia | | Singapore | |
|---|---|---|---|---|---|---|
|  | **Before** | **After** | **Before** | **After** | **Before** | **After** |
| Node | 165 | 162 | 94 | 86 | 42 | 30 |
| Edge | 12346 | 754 | 3668 | 227 | 798 | 46 |
| Density | 0.912 | 0.0578 | 0.839 | 0.0621 | 0.927 | 0.1057 |
| Average Degree | 149.648 | 9.309 | 78.043 | 5.279 | 38 | 3.067 |

Network analysis provides tools and metrics to evaluate the structural characteristics of news-media network. Node degree, the number of edges connected to a node, is a node-level measure that is used to assess the centrality of a node in the network [35]. In the context of news-media landscape, the degree of a node becomes a proxy to evaluate the diversity of people consuming a given media [18,20]. For example, if node A has strong relation with five other nodes, and node B has strong relation with only one other, we can infer that media A attracts people with a wider range of interests than media B. This research found that news outlets with the largest node degrees are Antara News in the Indonesian media network, Free Malaysia Today in the Malaysian media network, and Yahoo Singapore in the Singapore media network

Figure 3 shows the top five online news media ranking based on the degree value and the number of Twitter followers. The node degree of a news outlet represents the number of other news outlets that share audiences with this outlet, while the number of followers represents the size of the audience of this outlet on Twitter. As shown in the Figure 3, the results of the two indicators are relatively different. News outlets with the largest audience in their respective countries, namely Detik in Indonesia (15,070,895 followers), Astro Awani in Malaysia (1,686,834 followers), and Straits Times in Singapore (1,008,205 followers), have no position in the top five media list based on degree of centrality. Furthermore, in Indonesian media networks, the list of the top five news outlets based on these two indicators does not intersect at all. Meanwhile, there are three news outlets in Malaysia and Singapore

that have a large readership as well as a high degree of centrality, namely The Star, Malaysia Kini, and Harian Metro for Malaysian media networks, and Straits Times, Channel News Asia, and The New Paper for Singapore media networks. This shows that the centrality of a news outlet on media networks is not always reflected through the size of their followers. However, the greater the followers of an outlet, the more likely the outlet is to share audience with other news outlets.

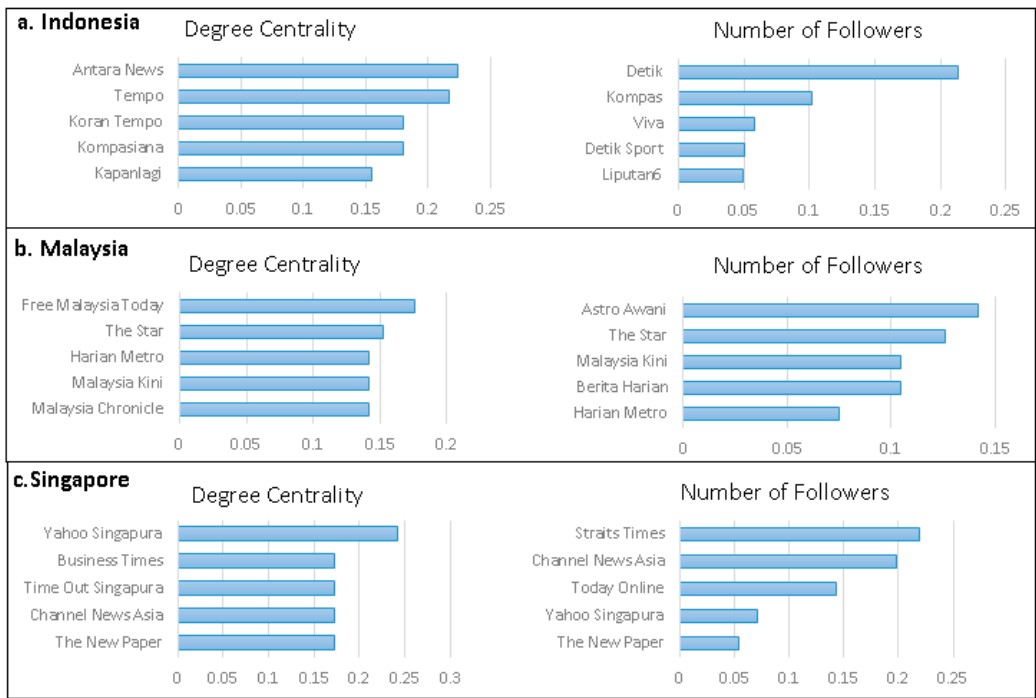

**Figure 3.** The top five online news media based on the degree value and the number of Twitter followers.

## 5. Discussion

The number of followers of a news outlet on Twitter gives us an overview of the size of the audience that outlets have on social media. However, in order to get a complete understanding of audience behavior and reveal the pattern of news consumption at the population level, we must view the news media landscape as a collection of interconnected news sources based on overlapping audiences. In this study, we used Twitter follower data to build online news media networks, where the nodes represent news outlets and the edges tell us how much overlap any two news outlets have in the audience they share. In other words, we wanted to map the media landscape, as it emerges from the preferences of social media users for news sources available online

Figures 4–6 show the visual representation of online news media networks in the three countries analyzed in this study. The networks were built using Gephi 0.91, an open source software for social networking analysis [36], with a list of edges between media outlets as input. By representing the landscape of online news media as a network, we can get a clear and measurable picture of the configuration of relationships between news outlets and the position of various news outlets in the media landscape.

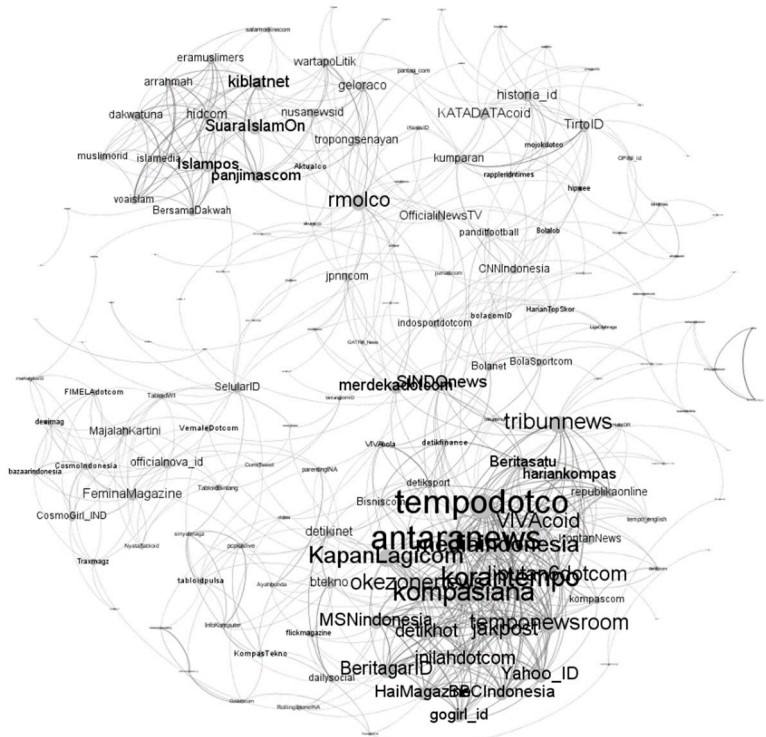

**Figure 4.** News media networks in Indonesia ($V_{indonesia}$ = 162, $E_{indonesia}$ = 754). Nodes represent online news media outlets and edges represent shared followers between any two outlets. The size and the label of a node is proportional to the degree centrality of the node.

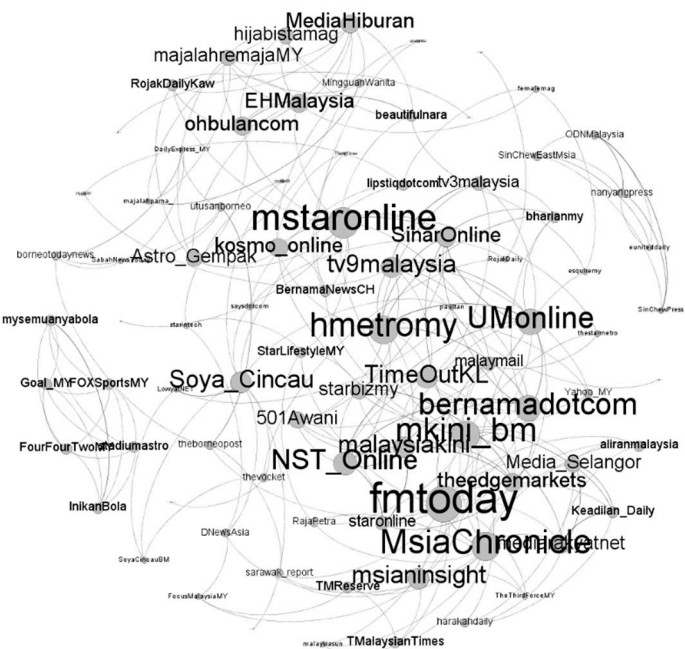

**Figure 5.** News media networks in Malaysia ($V_{malaysia}$ = 86, $E_{malaysia}$ = 227). Nodes represent online news media outlets and edges represent shared followers between any two outlets. The size and the label of a node is proportional to the degree centrality of the node.

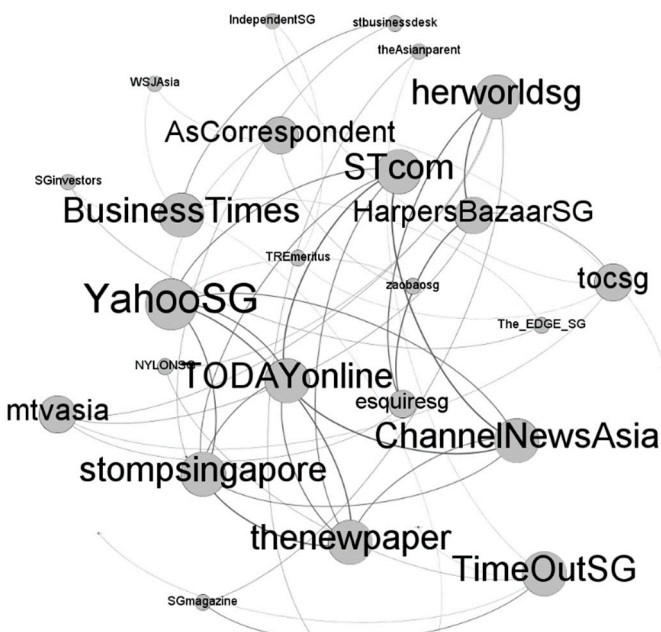

**Figure 6.** News media networks in Singapore ($V_{singapore} = 30$, $E_{singapore} = 46$). Nodes represent online news media outlets and edges represent shared followers between any two outlets. The size and the label of a node is proportional to the degree centrality of the node.

There are several ways to build a news media network based on follower overlap data. We can measure the similarity between news outlets using phi coefficients, as done in this study and in [19,21], or using closeness metrics [10,24], and then represent all the ties between news outlets collectively as undirected weighted networks [19,21,24] and directed weighted networks [10]. However, we argue that the resulting media network can look very different depending on how we filter the insignificant edges in the network. A weighted graph can be easily reduced to a sub-graph in which any of the edges' weights are larger than a given threshold. This global weight threshold technique has been applied in [19] using the *t*-test and in [10] using simple strategy: choose the strongest one. The short coming of this method is that it overpasses the nodes with small strength. Meanwhile, other studies analyze media networks as they are [24]. This can lead to wrong conclusions when carrying out network analysis given that the media networks constructed in these studies are correlational networks which naturally have a dense structure. In the analysis of media networks, network density needs to be reduced first to bring out the actual structure so that the media network can be analyzed reliably. Furthermore, in media networks, as shown in [19,21], both strength and weight distribution in general follows heavy tailed distribution which spans several orders of degrees. Applying a simple cutoff on weight will remove all the information below the cut-off. In this study, we show the effectiveness of the disparity filter method to extract the relevant connection backbone in media networks. Compared to other studies that use *t*-tests [19], the density of the final media networks in this study, as in [19], is very low which reflects this method can sufficiently reduce the network density without destroying its multi-scale nature.

After we represent the media landscape as a network, we can use the various metrics available in network analysis to summarize what the network reveals about news consumption. In this study, we applied metric degrees centrality to measure how central the position of news outlets is in the network. Traditionally, the size of the audience has been a measure of the prominence of a media. However, in this study we show that how strategic the position of a news outlet is in the network is not always reflected in the number of followers they have. In fact, compared to the size of their followers, media outlets reach a considerably larger audience through indirect exposure via social links [10]. An et al. [10] show that, based on indicators of social network exposure, a number of unconventional

news sources made up the top list media, along with several established media sources with millions of direct followers. In Figure 4, how central the position of a media outlet in the network is visualized by the size of the node, which is proportional to the value of the centrality of the media: the larger the node size, the more central the media represented by that node.

So far, the intersection between social media and news media mainly occurs on Facebook and Twitter. However, these two platforms are used for vastly different purposes when it comes to news consumption, due to the various features of the respective sites. Research on social media shows that consumers use Twitter for breaking news or searching information about their interest, but this is not the case for Facebook. Facebook members want to stay connected with their offline social network, and as such, tend to get news about their social environment through those networks [37]. However, news consumption on social media is strongly affected by the tendency of users to limit their exposure to a few sites [37]. That is why the structural characteristics of the news media landscape on the two platforms are relatively the same [38].

The news media network on Twitter presented in this study should be evaluated with consideration of two limitations. First, not all followers of news media accounts on Twitter are true audiences. It is known that about 15% of Twitter accounts are bot accounts that are controlled by software [39]. Second, not all news media followers are active users. Future research work should consider building a news media network based on followers who actively interact with news media accounts.

## 6. Conclusions

In this study, we used network analysis to represent and analyze the landscape of online news media in the social media environment based on follower overlap data. For this purpose, we developed the implementation stages of the proposed method, starting from data collection, modeling to data analysis, and applied the methodology to online news media data in three countries, namely Indonesia, Malaysia, and Singapore. In the data collection stage, we showed a long-tailed distribution pattern on the size of the media audience on Twitter. This indicates that the news media in the three countries analyzed is very heterogeneous in terms of the size of the audience, where there are a small number of news outlets with very large Twitter followers, while the majority of news outlets have only a few followers. This finding confirms a classic argument in media studies that public attention tends to be concentrated in a handful of news outlets. In the modeling stage, we showed the need for edge removal so that the news media network can be analyzed reliably. Comparison of network statistics before and after the elimination of non-significant edges indicates edge removal has an impact on the structure of networks. In the analysis phase, we used network indicators to evaluate the structural characteristics of the news media networks and show the ranking of news media outlets in the three countries analyzed based on the diversity of their audiences. In general, this study shows that the use of network analysis on follower overlap data can offer relevant insights about media diet and the way audiences navigate various news sources available on social media. Future work of this paper concerns deeper analysis of fragmentation phenomena in the pattern of news consumption on social media.

**Author Contributions:** On this research articles there are three contributors with these contributions: F.L.G. has been worked with the conceptualization and methodology on the data modeling. A.M. has been worked with the writing of the paper, formal analysis and validation of the experiments result and T.M. provided with the writing—review and editing of the manuscripts as well as supervision, project administration, and funding acquisition.

**Funding:** This research was funded by Japan Science and Technology Agency (JST) under Core Research for Evolutionary Science and Technology, Strategic Basic Research Program.

**Acknowledgments:** This research is supported by Department of Information Systems Management, Bina Nusantara University.

**Conflicts of Interest:** The authors declare no conflicts of interest.

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
