# Peer review of "Network Model for Online News Media Landscape in Twitter"

_information, doi:10.3390/info10090277_

Round 1

Reviewer 1 Report

The work introduces the adoption of Twitter data (and especially Twitter followers) in order to build online news media networks that can represent the pattern of news consumption in Twitter. Concretely, authors adopted a previous weighted network model and implemented the Filter Disparity Method with the aim of identifying the most significant overlaps in the network. Their experiments were based on news media outlets data from three countries, namely Indonesia, Malaysia and Singapore, and show that network analysis of followers’ data can offer relevant insights about media diet and the way readers navigate various news sources available on social media.

Although the problem of knowledge extraction in social networks is considered as a very challenging topic and moreover is relevant in the context of information, however the submission does not introduce any novel ideas regarding the execution of categorization in that context. The authors need to clarify and substantiate their contribution before the submission becomes worth publishing. The research appears not to have any in depth contributions.

Initially, the results are not adequate enough as authors need to differentiate their work from other authors’ contributions. Previously, authors need to better identify previous works in the corresponding area.

What is more, I think that the technical content could be enriched with use of the resulting pseudocode of the proposed schema.

Furthermore, it will be much advised for the authors to provide details regarding their implementation, e.g. for downloading tweets, extracting features, etc?

It is stressed that this work is based on twitter data. Is there any qualitative difference between twitter and facebook from an abstract perspective?

Aside that, the paper is not very well written; there are many typos and omissions.

There are also many recent contributions that authors should incorporate in terms of different methodologies. I suggest the authors to use some of these because it will be very interesting to see a comparison of their work with the others. Please also support your claims and discuss correlations of your proposed techniques with what has already been implemented in related work.

Finally, after reading the paper, it was not clear for me what was the "take away" message the work is attempting to convey as this contribution is considered as a journal and not a conference paper.

To sum up, I don't believe the paper is (yet) ready for publication.

Author Response

Response to reviewers’ comments

Title: Network Model for Online News Media Landscape in Twitter

Author: Firstname Lastname 1, Ford Lumban Gaol 2,* and Ardian Maulana 3

1   Affiliation 1; e-mail@e-mail.com

2   Computer Science Department, BINUS Graduate Program – Doctor of Computer Science, Bina Nusantara University, Jakarta 11480, Indonesia

3  Information Systems Management Department, Binus Graduate Program, Bina Nusantara University, Jakarta 11480, Indonesia; ardian.maulana@binus.ac.id

*  Correspondence: fgaol@binus.edu

Thank you for giving us the opportunity to submit a revised draft of my manuscript titled Network Model for Online News Media Landscape in Twitter  to Information Journal.We appreciate the time and effort that you and the reviewers have dedicated to providing your valuable feedback on our manuscript. We are grateful to the reviewers for their insightful comments on my paper. We wish to express our appreciation to the Reviewers for their insightful comments, which have helped us significantly to improve our manuscript. According to the suggestions, we have thoroughly revised our manuscript and its final version is enclosed.

Here is a point-by-point response to the reviewers’ comments and concerns.

Reviewer 1:

Although the problem of knowledge extraction in social networks is considered as a very challenging topic and moreover is relevant in the context of information, however the submission does not introduce any novel ideas regarding the execution of categorization in that context. The authors need to clarify and substantiate their contribution before the submission becomes worth publishing. The research appears not to have any in depth contributions.

Response: Thank you for your comment and suggestion. We agree with this and we have incorporated your suggestion throughout the revised manuscript. We have made some modification in section 1 to clarify the contribution of this paper. In the revised manuscript, the introduction section is written with the following structure:

Background information: page 1 line 24-28 Problem statement: page 29-33 Objective: page 1 line 40-44 Contribution: page 2 line 58-66

Change:

We have added a couple new paragraph to revised manuscript in page 1 line 29-33 and page 2 line 58-66. We have moved the paragraph in page 2 line 46-58 (previous version)  to  section 2: Related Work,  in revised version

Initially, the results are not adequate enough as: authors need to differentiate their work from other authors’ contributions.

Response: Thank you for your suggestion. We have added a couple of  sentences/paragrapgh throughout the revised manuscript to clarified the differentiation of our  work to other authors  

Change:

To distinguish our work with references [17] and references [19] in terms of: Methodology: in page 2 section 1 line 52-54 Data: in page 2 section 1 line 3 We distinguish our work with references [10] and references [20] in terms of: Methodology: in page 3 section 2 line 115-120

Previously, authors need to better identify previous works in the corresponding area.

Response:  Thank you for your suggestion. We have added new section, section 2: related work, to incorporate your suggestion in revised manuscript.

Change:

We have added new section: section 2 related work in page 2-3 line 74-120. What is more, I think that the technical content could be enriched with use of the resulting pseudocode of the proposed schema.

Response: We agree with your suggestion. We have added four   pseudocode  of the proposed schema in section 3  in revised manuscript.

Change:

We have added pseudo code for follower extraction process in page 4 section 3.1 line 46-47 We have added pseudo code for similarity measurement in page 6 line 194-195 We have added pseudo code for network construction in page 6 line 196-197 We have added pseudo code for edge filtering in page 7 line 216- 217

Furthermore, it will be much advised for the authors to provide details regarding their implementation, e.g. for downloading tweets,extracting features, etc?

Response: We agree with this and we have incorporated your suggestion throughout section 3.

Change:  We have added a couple new sentences in:

Page 4 section 3.1: data extraction line 137-139 Page 5 section 3.2.2: The construction of news media networks, line 190-193 Page 6-7 section 3.2.3: Edge Filtering, line 200-215

It is stressed that this work is based on twitter data. Is there any qualitative difference between twitter and facebook from an abstract perspective?

Response: We agree with this and we have added  new sentences/paragraph in  section 5: discussion to discuss the  difference between Twitter and Facebook in term of news consumption.

Change:

We have added new paragraph in page 12 section 5 line 351-359

There are also many recent contributions that authors should incorporate in terms of different methodologies. I suggest the authors to use some of these because it will be very interesting to see a comparison of their work with the others. Please also support your
claims and discuss correlations of your proposed techniques with what
has already been implemented in related work.

Response: Thank you for your suggestion. We have incorporated your suggestion in section 4-5  in revised manuscript.

Change:

To support our finding about follower distribution: we have added a couple new sentences in page 7-8  section 4 line 237-242.  To support our finding about media centrality: we have added a couple news sentences in page 11 section 5 line 342-348. To discuss contributions of other authors related to the construction of news media network and edge filtering method:  we have added  new paragraph in page 11 section 5 line 319-338.

Aside that, the paper is not very well written; there are many typos and omissions.

Response:

Response: The paper is checked again to remove the typo and omissions

Reviewer 2 Report

This is an interesting paper. In the paper, the authors used twitter follower data to build an online news media networks.  Then they identify the most significant overlaps in the network by implementing a weighted network model and the Filter Disparity Method. Their study shows that the use of network analysis on followers overlap data can offer relevant insights about media diet and the way audiences navigate various news sources available on social media.

The paper has good readability. It is easy to follow and understand. The feedback for improvement:

1.                 The paper’s structure could be improved. Instruction section gave the background information, problem statement, objectives, and contributions.

2.                 Add a related work section. The work done by other researchers using other methods should be discussed. 

3.                 Just wondering if the authors can give the justification of using disparity filter method for Edge Filtering and provide a bit more details about disparity filter method developed by others in section 2.2.3.

4.                 Is there any limitation in this study?

This is an interesting paper. In the paper, the authors used twitter follower data to build an online news media networks.  Then they identify the most significant overlaps in the network by implementing a weighted network model and the Filter Disparity Method. Their study shows that the use of network analysis on followers overlap data can offer relevant insights about media diet and the way audiences navigate various news sources available on social media.

The paper has good readability. It is easy to follow and understand. The feedback for improvement:

1.                 The paper’s structure could be improved. Instruction section gave the background information, problem statement, objectives, and contributions.

2.                 Add a related work section. The work done by other researchers using other methods should be discussed. 

3.                 Just wondering if the authors can give the justification of using disparity filter method for Edge Filtering and provide a bit more details about disparity filter method developed by others in section 2.2.3.

4.                 Is there any limitation in this study?

Author Response

Response to reviewers’ comments

Title: Network Model for Online News Media Landscape in Twitter

Author: Firstname Lastname 1, Ford Lumban Gaol 2,* and Ardian Maulana 3

1   Affiliation 1; e-mail@e-mail.com

2   Computer Science Department, BINUS Graduate Program – Doctor of Computer Science, Bina Nusantara University, Jakarta 11480, Indonesia

3  Information Systems Management Department, Binus Graduate Program, Bina Nusantara University, Jakarta 11480, Indonesia; ardian.maulana@binus.ac.id

*  Correspondence: fgaol@binus.edu

Thank you for giving us the opportunity to submit a revised draft of my manuscript titled Network Model for Online News Media Landscape in Twitter  to Information Journal.We appreciate the time and effort that you and the reviewers have dedicated to providing your valuable feedback on our manuscript. We are grateful to the reviewers for their insightful comments on my paper. We wish to express our appreciation to the Reviewers for their insightful comments, which have helped us significantly to improve our manuscript. According to the suggestions, we have thoroughly revised our manuscript and its final version is enclosed.

Here is a point-by-point response to the reviewers’ comments and concerns.

Reviewer 2:

The paper’s structure could be improved. Introduction section gave the background information, problem statement, objectives, and contributions.

Response: Thank you for your suggestion. We agree with this and we have modified the structure of the introduction section based on your suggestion as follow:

Background information: page 1 line 24-28 Problem statement: page 29-33 Objective: page 1 line 40-44 Contribution: page 2 line 58-66

Change:

We have added a couple new paragraph to revised manuscript in page 1 section 1 line 29-33 and page 2 section 1 line 58-66. We have moved the paragraph in page 2 line 46-58 (previous version) to  section 2: Related Work,  in revised version Add a related work section. The work done by other researchers using other methods should be discussed.

Response: Thank you for your suggestion. We agree with this suggestion and we have added section 2: related work to our revised manuscript

Change:

We have added new section: section 2 Related Work, in page 2-3 line 74-120.

Just wondering if the authors can give the justification of using disparity filter method for Edge Filtering and provide a bit more details about disparity filter method developed by
others in section 2.2.3.

Response: Thank you for your suggestion. We have incorporated your suggestion in our revised manuscript.

Change:

We have added a couple of sentences in page 6-7 section 2.3 line 200-215

Is there any limitation in this study?

Response: Thank you for pointing this out. We have incorporated your suggestion in our revised manuscript.

Change:

We have added new paragraph in page 12 section 5 line 360-365

Round 2

Reviewer 1 Report

The authors have cleared most of my concerns. Please update the name of the first author.

Overall, this is an interesting concept and has been shown to be successful for the behavior of news consumption in social media environment. However, I believe that this paper lacks a concise overview of why this is important and how it builds upon similar articles published in the last days.

So authors can incorporate some recent contributions (in alphabetic order) in terms of messages diffusion in Twitter:

Kafeza et al., T-PCCE: Twitter Personality based Communicative Communities Extraction System for Big Data. IEEE Transactions on Knowledge and Data Engineering (TKDE), 2019. Kafeza et al., Predicting Information Diffusion Patterns in Twitter. Artificial Intelligence Applications and Innovations (AIAI), 2014.

Author Response

Response to reviewers’ comments

Title: Network Model for Online News Media Landscape in Twitter

Ford Lumban Gaol1,*, Tokuro Matsuo 2, and Ardian Maulana 3

1   Computer Science Department, BINUS Graduate Program – Doctor of Computer Science, Bina Nusantara University, Jakarta 11480, Indonesia

2   Information System Department, Advanced Institute of Industrial Technology, Tokyo, Japan;

3  Information Systems Management Department, Binus Graduate Program, Bina Nusantara University, Jakarta 11480, Indonesia; ardian.maulana@binus.ac.id

*   Correspondence: fgaol@binus.edu

Thank you for giving us the opportunity to submit a revised draft of the manuscript titled Network Model for Online News Media Landscape in Twitter  to Information Journal. We appreciate the time and effort that you and the reviewers have dedicated to providing your valuable feedback on our manuscript. We are grateful to the reviewers for their insightful comments on the paper. We wish to express our appreciation to the Reviewers for their insightful comments, which have helped us significantly to improve our manuscript. According to the suggestions, we have thoroughly revised our manuscript and its final version is enclosed.

Here is a point-by-point response to the reviewers’ comments and concerns.

Overall, this is an interesting concept and has been shown to be successful for the behavior of news consumption in social media environment. However, I believe that this paper lacks a concise overview of why this is important and how it builds upon similar articles published in the last days. So authors can incorporate some recent contributions (in alphabetic order) in terms of messages diffusion in Twitter:

Response: Thank you for your suggestion. We agree with this and we have incorporated your suggestion in introduction section.

Change:

We have added a sentences in page 1 line 43-44 We have added new references, that is references 11 and 12, in line 417-421.
